# The Effect of Xanthohumol Derivatives on Apoptosis Induction in Canine Lymphoma and Leukemia Cell Lines

**DOI:** 10.3390/ijms241411724

**Published:** 2023-07-21

**Authors:** Małgorzata Grudzień, Aleksandra Pawlak, Tomasz Tronina, Justyna Kutkowska, Angelika Kruszyńska, Jarosław Popłoński, Ewa Huszcza, Andrzej Rapak

**Affiliations:** 1Department of Experimental Oncology, Ludwik Hirszfeld Institute of Immunology and Experimental Therapy, Weigla 12, 53-114 Wroclaw, Poland; malgorzata.grudzien@hirszfeld.pl (M.G.); justyna.kutkowska@hirszfeld.pl (J.K.); angelika.kruszynska@hirszfeld.pl (A.K.); 2Department of Pharmacology and Toxicology, Faculty of Veterinary Medicine, Wroclaw University of Environmental and Life Sciences, Norwida 31, 50-375 Wroclaw, Poland; aleksandra.pawlak@upwr.edu.pl; 3Department of Chemistry, Wroclaw University of Environmental and Life Sciences, Norwida 25, 50-375 Wroclaw, Poland; tomasz.tronina@upwr.edu.pl (T.T.); jaroslaw.poplonski@upwr.edu.pl (J.P.); ewa.huszcza@upwr.edu.pl (E.H.)

**Keywords:** hop flavonoids, xanthohumol, biotransformation, canine cell lines, apoptosis induction, cancer treatment

## Abstract

Xanthohumol is a cancer chemopreventive agent that can interfere with the initiation, promotion, and progression phase of carcinogenesis via a variety of inhibitory mechanisms. Xanthohumol was reported as an effective agent against leukemia/lymphoma cells. In the present study, we investigated the effect of xanthohumol and its natural and semisynthetic derivatives against various canine leukemia/lymphoma cell lines. Xanthohumol, three hops minor prenylflavonoids (xanthohumol C, xanthohumol D, α,β-dihydroxanthohumol) and four derivatives obtained by biotransformation (xanthohumol 4′-O-β-D-(4‴-O-methyl)-glucopyranoside) as well as by chemical modification (1″,2″-dihydroxanthohumol K, 2,3-dehydroisoxanthohumol, (Z)-6,4′-dihydroxy-4-methoxy-7-prenylaurone) were tested for their antiproliferative and pro-apoptotic activities against the following canine leukemia/lymphoma cell lines: CLBL-1 (B-cell lymphoma), CLB70 (B-cell leukemia), and GL-1 (B-cell leukemia). The compounds were tested at a final concentration range of 0.1–30 µM for 48 h. All eight of the tested flavonoids exerted concentration-dependent cytotoxicity in the selected canine lymphoma/leukemia cell lines. Three compounds markedly decreased the viability of all cell lines with IC_50_ in the range of 0.5 to 8 μM. Double-staining of the treated cells with AnnexinV and propidium iodide revealed that the dying cells were mostly in the late apoptosis stage. ROS production and changes in mitochondrial potential were detected. Western blot analysis showed a decreased expression of Bcl-2. Canine lymphoma and leukemia cell lines are sensitive to xanthohumol derivatives, and the compounds acted through an apoptotic cell-death mechanism. These compounds, either used alone or in combination with other therapies, may be useful for the treatment of canine leukemia/lymphoma.

## 1. Introduction

According to estimates from the World Health Organization (WHO), cancer was the second-leading cause of death globally in 2018. Non-Hodgkin lymphoma (NHL) accounts for about 4% of all cancers [1]. The main cause of dog mortality, apart from infectious diseases, is also cancer. Lymphomas constitute about 90% of all hematopoietic neoplasms found in dogs, which in turn constitute about 25% of all diagnosed types of cancer in this animal species [2]. The incidence of lymphomas in the canine population is higher than in humans. Some of them, such as non-Hodgkin lymphoma (NHL) and lymphocytic leukemia, are very similar in the etiology, pathogenesis, and response to treatment to the diseases occurring in humans. Due to their anatomical and physiological similarities to humans, dogs are a useful model for the study of new therapeutic strategies for humans.

There are many different approaches to treating cancers; however, they are often painful because of undesirable side effects or are ineffective due to the cancer cells’ resistance to chemotherapeutics. These problems result in a steady increase in the demand for the development of new drugs. The discovery of new anticancer agents based on natural products are a focus of much research because many phytochemicals have demonstrated selective cytotoxicity towards various types of human cancer cells, accompanied by a minimal toxicity to normal cells [3].

Of the 175 approved small-molecule anticancer agents available for use (since 1940 to the end of 2014) 131 are other than synthetic, and 85 are either natural products or are directly derived therefrom [4]. Among phytochemicals, flavonoids have attracted much attention due to their exceptional spectrum of pharmacological activities and because they are essential components in human and animal diets, as has been summarized in numerous comprehensive review articles [5,6].

One of the most important and therefore most frequently studied properties of flavonoids is their antitumor activity. Epidemiological and clinical studies indicate that flavonoids can prevent carcinogenesis as well as suppress the growth of established tumors [7,8,9,10,11].

Xanthohumol is the most abundant prenylated flavonoid in hops (0.1–1%, *w*/*w*) accompanied by related chalcones and flavanones, all of which occur at 10–100-fold lower concentrations [12]. Hop cones, due to their high content of bitter acids (humulones and lupulones) and essential oils, are a difficult matrix for isolating flavonoids. A much cheaper and easier to process source for obtaining prenylated hop flavonoids is spent hops—a byproduct formed after the selective isolation of bitter acids from hop cones using supercritical carbon dioxide. However, due to the low flavonoid content related to the variability in hop varieties, and their potential degradation due to the poor storage of spent hops, as well as the time-consuming complex isolation process which requires specialized equipment and involves coupled extraction with organic solvents and purification by chromatographic methods (using Sephadex-LH20 and silica gel), it is still challenging to obtain large amounts of xanthohumol and other natural prenylated flavonoids. Nevertheless, the relative ease of access to spent hops as well as their low cost (as a brewery industry by-product), prenylated hop flavonoids have become the objects of numerous studies focused on their use in cancer prevention and treatment.

Xanthohumol is a broad-spectrum cancer chemopreventive agent that can interfere with the initiation, promotion and progression phase of carcinogenesis via a variety of inhibitory mechanisms [13,14]. It has been shown that xanthohumol modulates the activity of enzymes related to cellular proliferation, differentiation, apoptosis, and inflammation as well as enzymes involved in carcinogen metabolism [13,14,15]. The very promising anticancer activity of xanthohumol has also led researchers to evaluate other prenylated hop flavonoids such as isoxanthohumol, a flavanone resulting from the cyclization of xanthohumol, and 8-prenylnaringenin, the strongest phytoestrogen known to date [15], and to their analogues obtained by chemical or enzymatic modifications [16,17]. The influence of xanthohumol on blood cancer cells was studied both in vitro and in vivo. Lu with coworkers showed that xanthohumol inhibits the proliferation, induces S phase cell-cycle arrest, stimulates the apoptosis and decreases the BCR-ABL oncoprotein level in the K562 cell line (human erythroleukemia) [18]. Xanthohumol also leads to cell death in HL-60 (human acute myeloid leukemia) cells by increasing the level of ROS and membrane permeability and causing LDH release and cell swelling [19]. Additionally, studies performed on B-cell acute lymphocytic leukemia cell lines showed that the antileukemic activity of xanthohumol is associated with impaired cell migration and invasion, which was confirmed also by in vivo testing [20].

In the present study we investigated the effect of xanthohumol and its natural and semisynthetic derivatives with a modified flavonoid skeleton (α,β-dihydrochalcone, flavanone, flavone, aurone) and/or a modified prenyl group (oxidized, cyclized) functionalized by glycosylation on anti-inflammatory, antiproliferative and proapoptotic activities against various canine leukemia/lymphoma cell lines. Today, one of the main causes of dog mortality, apart from infectious diseases, is cancer. Lymphomas constitute about 90% of all hematopoietic neoplasms found in dogs, which in turn constitute about 25% of all diagnosed types of cancer in this animal species. The incidence of lymphomas in the canine population is higher than that in humans. Some of them, such as non-Hodgkin lymphoma (NHL) and lymphocytic leukemia, are very similar in the etiology, pathogenesis, and response to treatment to the diseases occurring in humans. Due to their anatomical and physiological similarities to humans, dogs are a useful model for the study of new therapeutic strategies for humans.

To the best of our knowledge, this is the first scientific report regarding the activity of prenylated hop flavonoids against canine cancer cell lines.

## 2. Results

### 2.1. Compounds Preparation

Xanthohumol (**1**) and their seven derivatives (Figure 1) were tested for their antiproliferative and proapoptotic activities against canine cancer cell lines. Xanthohumol C (**2**), Dihydroxanthohumol K (**6**) and Dihydroxy-4-methoxy-7-prenylaurone (**8**) were obtained by cyclization of xanthohumol (**1**), whereas 2,3-Dehydroisoxanthohumol (**7**) was obtained from isoxanthohumol, which is formed in the chalcone-flavanone cyclization reaction of xanthohumol (**1**) in alkaline environment, α,β-Dihydroxanthohumol (**4**) was prepared from xanthohumol (**1**) by regioselective hydrogenation reaction. Xanthohumol 4′-O-β-D-(4‴-O-methyl)-glucopyranoside (**5**) was obtained as a product of the microbial transformation of xanthohumol (**1**). These compounds were obtained earlier and described by our group [16,17,21,22,23]. In this article, we describe a newly developed simple method for the separation of xanthohumol D (**3**) from spent hops connected with the simultaneous purification of xanthohumol (**1**), which, as a prenylated flavonoid found in hops in quantities up to 1%, *w*/*w*, [12], is usually the main focus of interest. The efficiency of the isolation process carried out was 2.01 g of 1 and 0.44 g of 3 per 1 kg of spent hops. Because column chromatography on Sephadex LH-20 is easy to scale up, a sufficient amount of purified minor hops flavonoid xanthohumol D (**3**) can be obtained to evaluate its bioactivity (Figure 1).

### 2.2. IC_50_ Measurement

We evaluated the IC_50_ value by treating three canine cancer cell lines with xanthohumol (**1**) and its derivatives **2**–**8** (Table 1). Xanthohumol (**1**) was used as a reference compound. The results revealed that CLB70 (B-cell leukemia) [24] and CLBL-1 (B-cell lymphoma) [25] cells were relatively sensitive for the applied compounds, whereas GL-1 (B-cell leukemia) cells [26] were the most resistant among the tested lines. Compounds **1**, **3** and **5** displayed the lowest IC_50_ values (Appendix A). Other tested compounds showed effective killing of cancer cells in concentrations higher than **1**, with the small exception of the IC_50_ value for **2** when applied to GL-1 cells; this concentration was similar to the concentration obtained for **1** applied to GL-1 cells (Table 1).

### 2.3. Evaluation of Apoptosis

Based on IC_50_ values, we have chosen **1**, **3** and **5** for further investigation. The analysis of apoptosis induced by **1** and its analogs in canine cancer cells showed that all three cell lines were susceptible to apoptosis induced by the applied flavonoids. The highest number of apoptotic cells was visible after **1** and **3** treatments. The apoptotic effect of **5** was smaller in CLBL-1 and GL-1 cells but similar to other tested compounds in the case of CLB70 cells (Figure 2).

### 2.4. Measurement of Mitochondrial Potential

In order to check whether xanthohumol (**1**) and its analogs influence on mitochondrial potential, we treated canine cancer cells with the investigated compounds for 24 h and measured the number of cells with decreased mitochondrial potential. Again, GL-1 line was the less sensitive for the treatment whereas CLB70 and CLBL-1 responded to the treatment by the increase of cells with decreased ΔΨ (Figure 3).

### 2.5. Analysis of ROS Production

Next, we decided to investigate the effect of **1**, **3** and **5** on ROS production in canine cancer cells. Compounds **1** and **3** were able to increase ROS production in CLBL-1 and CLB70 cells. GL-1 cells did not respond to any of the investigated substances. Xanthohumol glucoside (**5**) was not effective in the induction of ROS production in any of the cell lines (Figure 4).

### 2.6. Expression of Antiapoptotic Proteins

Treatment with **1**, **3** and **5** in concentration 3 μM each for 48 h caused decreased expression of Bcl-2 in CLBL-1 and CLB70 cell lines, especially when treated with xanthohumol. The expression of ERK kinases is unchanged and PARP protein digestion is barely detectable (Figure 5). 

## 3. Discussion

Comparative oncology is a rapidly evolving discipline aimed at integrating research on naturally occurring cancer in animals and applying it to more general research in cancer biology and therapy. Among the various species of animals, the dog is the most appropriate in terms of its body size, life expectancy and shared living conditions with humans.

Due to their anatomical and physiological similarities, dogs have become a useful model for research into new drugs and treatments for humans. Spontaneously emerging tumors in dogs also share a number of clinical and molecular features with human tumors. It has also been suggested that canine cell lines could become a good research model for the testing of new therapeutic strategies in the treatment of human leukemias and lymphomas. The fight against cancer requires different therapeutic approaches to minimize side effects for patients. Naturally derived compounds and their analogs appear to be helpful either in combination with chemotherapeutics as we showed in [27], or as anticancer agents [10]. Xanthohumol (**1**), a flavonoid isolated from spent hops, is easily accessible and displays promising anticancer activity [14]; therefore, we decided to investigate the properties of **1** and its several derivatives using canine cancer cells. Dogs, like humans, also suffer from cancer [28] and they are used as a model for studying drug toxicology; however, we did not find any reports focusing on the influence of **1** on canine cancer cells, therefore we decided to conduct this type of research. The similar clinical picture of the disease, as well as the response to therapy, enables the use of studies conducted in dogs to not only monitor the effectiveness of new therapeutic strategies, but to also observe and attempt to reduce the side effects of some drugs.

We isolated two chalcones, xanthohumol (**1**) and xanthohumol D (**3**), from a waste material, spent hops. Then, using **1** as a starting material, we obtained another six xanthohumol (**1**) derivatives by bio- or chemical transformations, which differed from xantohumol (**1**) in their flavonoid skeleton, modification in the prenyl group, and the presence of a sugar derivative, which in the case of flavonoids can affect their better absorption. Therefore, the goal of this study was to determine: (1) the effect of the type of flavonoid skeleton (chalcone **1**, α,β-dihydrochalcone **4**, flavone **7**, aurone **8**); (2) modification in the prenyl group (cyclization—compounds **2** and **6**, hydroxylation—compound **3**); and (3) the presence of a sugar moiety (compound **5**) on the antiproliferative activities against three canine cancer cell lines: CLBL-1, CLB70 and GL-1. The study showed that changing the skeleton of the prenylated flavonoid from chalcone to another, as well as cyclization of the prenyl group, reduced the antiproliferative activity against the tested cancer cell lines. The presence of a glycosylated derivative of xanthohumol **5** among the tested compounds is not accidental. The biological activity of many compounds including flavonoids may be limited by their low solubility in water, which may affect their uptake. Pharmacokinetic study on rats which orally administered doses of 40, 100, and 200 mg xanthohumol (1) kg-1 body weight confirmed the low absolute bioavailability (1.16%, 0.96%, and 0.53%, respectively [29], which may be the result of very limited uptake in the small intestine and rapid metabolization by microorganisms in the colon [12]. The presence of a sugar moiety in the flavonoid molecule was proposed as the crucial determinant of its absorption in humans [30]. A study performed on Beagle dogs showed that bioavailability of 3-O-glycoside of quercetin (isoquercetin) was higher (235%) in comparison to its aglycone quercetin [31]. Therefore, we also decided to determine the activity of xantohumol glucoside 5, which may increase the bioavailability of xanthohumol, against the tested canine cancer cell lines.

The measurement of IC_50_ towards canine cancer cell lines allowed us to choose chalcone **1** and its two analogs: **3** and **5** for further studies due to their lower IC_50_ values. Xanthohumol C (**2**) was less effective than 1, in contrast to the results obtained by Roehrer et al. [23]. It should come as no surprise that the compounds act differently on different cell lines (dog blood cancer and human breast cancer). Among our tested cells, the GL-1 cell line appeared to be the most resistant to the treatment, which was also visible during other experiments in this project as well as in our previous findings [32]. 

The analysis of apoptosis revealed that **1**, **3** and **5** treatments caused an increase in apoptotic cell numbers, and in this case **1** and **3** were shown to be more effective than **5**. These three compounds, in terms of their chemical structure, are characterized by a chalcone skeleton and uncyclized prenyl group, which are the only noteworthy structure-activity motifs observed in the tested panel of compounds.

Our results regarding the influence of **1** on cancer cells are consistent with the findings of Deeb and colleagues [33] who revealed that xanthohumol (**1**) increased the number of apoptotic prostate cancer cells when applied in the concentration range 5–40 μM. The analysis of ROS production showed that case **5** was also less efficient in causing the desired effect in comparison to **1** and **3**. We were able to detect a decrease in the expression of the anti-apoptotic protein Bcl-2 in the two tested B-type lines. The Bcl-2 protein belongs to the Bcl-2 protein family and acts as pro-survival protein and can inhibit apoptosis. Overexpression of Bcl-2 protein is associated with a poor prognosis during tumour treatment. The CLBL-1 line is a representative of DLBCL lymphoma, which is most common in dogs. The significant reduction in Bcl-2 expression in this line after treatment with xanthohumol derivatives is beneficial. The same compounds can induce different proteins and signaling pathways in different cell lines. A precise elucidation of the mechanism of action would require examining the expression of many other signaling proteins.

## 4. Materials and Methods 

### 4.1. Reagents

Methanol and acetone of analytical grade, dimethyl sulphoxide (DMSO) and ethanol were purchased from POCh (Gliwice, Poland). Methanol of HPLC grade was purchased from Merck (Darmstadt, Germany). Sephadex LH-20, formic acid of analytical grade, fetal bovine serum, L-glutamine, penicillin and streptomycin solutions, thiazolyl blue tetrazolium bromide (MTT), sodium dodecyl sulfate (SDS), and N,N-dimethylformamide (DMF) were purchased from Sigma-Aldrich (Chemie GmbH, Steinheim, Germany). Phosphate-buffered saline (PBS) and an RPMI 1640 culture medium were obtained from the Institute of Immunology and Experimental Therapy, Wroclaw. Water for all experiments was provided by a Milli-Q water purification system from Millipore.

The spent hops were produced in the New Chemical Syntheses Institute, Puławy, Poland by supercritical carbon dioxide extraction according to [34]. Cones (hops) of *Humulus lupulus* cv. ‘Magnum’ collected in 2015 in Lublin region (SE Poland) were used in the extraction process. Flavonoids used in these studies were isolated from spent hops or synthesized. The protocols for obtaining these compounds and their spectroscopic data were reported in previous studies from our group (see Section 4.2).

### 4.2. Preparation of Flavonoids

Xanthohumol (**1**) and xanthohumol D (**3**) were isolated from spent hops. Plant material (250 g) was extracted with acetone (1 L) for 24 h at room temperature on a rotary shaker. After solvent removal under a vacuum, the concentrated extract (7.45 g) was further dissolved in 25 mL of methanol and filtrated through filter paper in order to dispose of difficult-to-dissolve ballast substances. The filtrate was concentrated in a vacuum (7.28 g) and then subjected to Sephadex LH-20 column chromatography using methanol as an eluent and a total of 62 (50 mL each) fractions were collected. Fractions 41–57 containing mainly xanthohumol (**1**) and xanthohumol D (**3**) were combined and then concentrated in a vacuum (1.99 g). 

The procedure described above was repeated four times (i.e., for 4 × 250 g of spent hops) to give in total 7.98 g of a crude mixture of 1 and 3, which was subjected repeatedly to separation on a Sephadex LH-20 column using methanol as the eluent. Xanthohumol D (**3**) of purity 95.6% was separated in fractions 41–44 (0.441 g) while xanthohumol (**1**) in the amount of 2.01 g of purity 96.8% was separated in fractions 47–56. Xanthohumol (**1**) and xanthohumol D (**3**) intended for use in biological studies were further purified to >98% by column chromatography on silica gel 60 sing methylene chloride:diethyl ether:hexane:formic acid (200:40:28:1 *v*/*v*) as the eluent to give yellow-orange crystals of both compounds **1** and **3**. The purity of the obtained flavonoids was determined with UHPLC. 

Xanthohumol C (**2**) was obtained by cyclization of xanthohumol (**1**) using 2,3-dichloro-5,6-dicyan-1,4-benzochinone (DDQ) in anhydrous 1,4-dioxane at 95 °C according to the Vogel and Heilman method [35] modified by Popłoński et al. [17]. α,β-Dihydroxanthohumol (**4**) was prepared from xanthohumol (**1**) by regioselective hydrogenation reaction with hydrogen gas, catalyzed with palladium (5%) on carbon [36]. Xanthohumol 4′-*O*-β-D-(4‴-*O*-methyl)-glucopyranoside (**5**) was obtained as the product of microbial transformation of xanthohumol (**1**) by fungi *Beauveria bassiana* [21]. 1″,2″-Dihydroxanthohumol K (**6**) was obtained by cyclization of xanthohumol (**1**) using trifluoroacetic acid [17]. 2,3-Dehydroisoxanthohumol (**7**) was obtained from isoxanthohumol, which is formed in the chalcone–flavanone cyclization reaction of xanthohumol (**1**) in alkaline environment [22]. Dehydrogenation of isoxanthohumol was performed with iodine-pyridine complex [17]. (*Z*)-6,4′-Dihydroxy-4-methoxy-7-prenylaurone (**8**) was prepared from xanthohumol (**1**) in a chalcone–aurone cyclization reaction using mercury acetate in pyridine [23].

### 4.3. Identification with UHPLC

HPLC was performed on an Ultimate 3000 UHPLC^+^ focused instrument (Thermo Scientific, Waltham, MA, USA) with a photodiode array detector (detection from 210 to 450 nm wavelength) using Acclaim RSLC PolarAdvantage II column (100 × 2.1 mm, 2.2 µm, Thermo Scientific, Waltham, MA, USA) at the flow rate of 0.8 mL/min and the following elution program: isocratic elution from 0 to 1 min (45% B); gradient elution: from 1 to 1.2 min (45% B → 50% B), from 1.2 to 1.6 min (50% B → 60% B), from 1.6 min to 4.4 min (60% B → 65% B), isocratic elution: from 4.4 to 5.2 min (65% B), gradient elution: from 5.2 to 6.2 (65% B → 100% B), isocratic elution: from 6.2 to 7.00 min (100% B), gradient elution: from 7.0 to 7.1 min (100% B → 45% B), then isocratic elution: from 7.1 to 8.0 min (45% B). As solvents, A 0.1% HCOOH in water, and B 0.1% HCOOH in MeCN were used. The temperature of the column during separation was set at 28 °C. The retention time (Rt) and wavelength (λ) of the obtained compounds are shown in Table 2.

### 4.4. Cell Lines and Cell Culture

The following canine cell lines: CLB70 (B-cell leukemia), CLBL-1 (B-cell lymphoma) and GL-1 (B/T-cell leukemia) were used in this study. The CLB70 cell line was obtained by Aleksandra Pawlak and coworkers [24]; CLBL-1 was obtained from Barbara C. Ruetgen, Institute of Immunology, Department of Pathobiology, University of Veterinary Medicine, Vienna, Austria [25]; and GL-1 cells were obtained from Yasuhito Fujino and Hajime Tsujimoto from the University of Tokyo, Department of Veterinary Internal Medicine [26]. 

CLB70, CLBL-1 and GL-1 cell lines were cultured in Advanced RPMI 1640 medium supplemented with 2 mM L-glutamine, 100 U/mL penicillin and 100 µg/mL streptomycin and 10% fetal bovine serum (FBS) (CLB70 and GL-1) and 20% FBS (CLBL-1). The culture was maintained at 37 °C in a 5% CO_2_ incubator in a humidified atmosphere. The cells were cultured in 75 cm^2^ (CLBL-1) and 25 cm^2^ (CLB70 and GL-1) flasks (Corning, Corning, NY, USA) and split every second day to maintain optimal density (50–70% of cell density).

### 4.5. Evaluation of Changes in Mitochondrial Potential

Changes in mitochondrial potential were detected using JC-1 Mitochondrial Membrane Potential Flow Cytometry Assay Kit (Cayman Chemicals, Ann Arbor, MI, USA) according to the manufacturer’s instructions. Briefly, the cells were treated with flavonoids **1**, **3** and **5** and valinomycin (as a positive control) for 24 h. After treatment, the cells were washed twice with PBS containing 2.5% FBS and incubated with JC-1 stain (5 μg/mL) suspended in culture media for 20 min in the dark at 37 °C. After that, the cells were washed again twice, resuspended in PBS containing 2.5% FBS and measured using FACS Calibur. A minimum of 10,000 gated events per tube were collected. The fluorescence of JC-1 dye monomers was detected in the FL-2 channel and the signal from the aggregates was identified in FL-1 channel. The results were analyzed in Flowing Software 2.5.1 (Perttu Terho, Turku, Finland).

### 4.6. Evaluation of Reactive Oxygen Species (ROS)

ROS production was detected using H2DCFDA reagent (Thermo Fisher Scientific, Waltham, MA, USA). Cells were treated with flavonoids **1**, **3** and **5** for 1 h in culture conditions, the culture media were removed, and cells were suspended in H2DCFDA reagent in concentration 10 μM, diluted in PBS containing 10% FCS. The cells were incubated in culture conditions for another hour, washed twice with PBS containing 10% FCS and incubated for another 30 min in culture conditions, followed by two washings with PBS. Pellets were suspended in PBS containing 2.5% FCS and analyzed using FACS Calibur. A minimum of 10,000 gated events per tube were collected. The fluorescence of reagent was identified in FL-1 channel. Cells incubated without the addition of investigated agents served as a control. 

### 4.7. Annexin V/PI Staining

After seeding at a density of 1 × 10^5^/mL in 96-well plates (TPP, Trasadingen, Switzerland) the cells were incubated for 48 h with tested concentration (3 or 10 μM) of xanthohumol (**1**) and its most active derivatives: xanthohumol D (**3**) and xanthohumol glucoside (**5**). Cells were then collected, suspended in a binding buffer, and stained with Annexin V-FITC and PI (final PI concentration 1 μg/mL). Flow cytometric analysis was immediately performed using a flow cytometer (FACS Calibur; Becton Dickinson, Biosciences, San Jose, CA, USA). CellQuest 3.l.f. software (Becton Dickinson, San Jose, CA, USA) was used for the data analysis.

### 4.8. MTT Test

The proliferation of canine cells was determined using the MTT test (Sigma Aldrich, Steinheim, Germany). In brief, 1 × 10^5^ cells per well were seeded in a 96-well-plate (Thermo Fisher Scientific, Denmark), and xanthohumol (**1**) and its derivatives were added in the concentration range of 0.1–30 μM. After incubation for 48 h, 20 μL of MTT solution (5 mg/mL) were added to each well. After dissolving of the contents, the optical density of the wells was measured with a spectrophotometric microplate reader (Elx800, BioTek, Winooski, VT, USA) at a reference wavelength of 570 nm. The results were calculated as IC_50_ (50% inhibitory concentration) from the graph plotting using Prism GraphPad 7.0 software. The values were determined as the means from three independent experiments (three wells each).

### 4.9. Western Blot Analysis

Cells were collected and lysed using cold RIPA buffer (150 mM NaCl (POCH), 50 mM Tris–HCl pH 8.0 (BioShop, Burlington, ON, Canada), 1% NP-40 (Calbiochem, San Diego, CA, USA), 0.5% sodium deoxycholate (Sigma-Aldrich, St. Louis, MO, USA), and 1% SDS (BioShop, Burlington, ON, Canada)) supplemented with SigmaFAST Protease Inhibitor Cocktail (Sigma-Aldrich, St. Louis, MO, USA) and Halt Phosphatase Inhibitor Cocktail (Thermo Fisher Scientific, Waltham, MA, USA). The lysates were sonicated for 10 s at 100% power using a Sonopuls HD 2070 ultrasonic homogenizer (Bandelin, Berlin, Germany) and centrifuged at 10,000× *g* for 10 min at 4 °C to pellet cellular debris. The protein concentration was established using Pierce BCA Protein Assay Kit (Thermo Fisher Scientific, Waltham, MA, USA), according to the manufacturer’s protocol. Absorbance at 570 nm was measured using a Wallac 1420 VICTOR2 plate reader. Cell lysates were then diluted in Laemmli sample buffer (50 mM Tris–HCl pH 6.8, 10% glycerol (BioShop, Burlington, ON, Canada), 5% 2-mercaptoethanol (Sigma-Aldrich, St. Louis, MO, USA), 2% SDS, 0.05% bromophenol blue (BioShop, Burlington, ON, Canada)) and heated for 5 min at 95 °C. The samples were then loaded on SDS-PAGE resolving gels (SDS-PAGE running buffer: 25 mM Tris, 192 mM glycine (BioShop Burlington, ON, Canada), 0.1% SDS) and transferred (semidry transfer) to a PVDF membrane (0.45 μm pore size; Merck Millipore, Burlington, MA, USA) (transfer buffer: 25 mM Tris, 192 mM glycine, and either 10% or 20% methanol (POCH)). In between the steps, membranes were washed with TBST (20 mM Tris, 150 mM NaCl, 0.1% Tween 20 (BioShop, Burlington, ON, Canada)). Membranes were blocked either with 1% casein (0.1 M Tris–HCl pH 8.0, 214 mM NaCl overnight at 4 °C and then incubated with the primary antibody overnight at 4 °C. After probing with HRP-conjugated secondary antibody for 1 h in RT, proteins of interest were detected using SuperSignal West Dura Extended Duration Substrate (Thermo Fisher Scientific, Waltham, MA, USA). The following antibodies were used in this study: anti-PARP (1:1000, sc-7150; Santa Cruz Biotechnology, Dallas, TX, USA), anti-Bcl-2 (1:1000, sc-7382; Santa Cruz Biotechnology), anti ERK (1:1000, sc-93; Santa Cruz Biotechnology), anti-actin/HRP (1:2000, sc-1615; Santa Cruz Biotechnology, Dallas, TX, USA), anti-mouse/HRP (1:2000, P0447; Dako, Glostrup, Denmark), and anti-rabbit/HRP (1:2000–3000, P0048; Dako, Glostrup, Denmark).

## 5. Conclusions

In conclusion, three natural flavonoids, xanthohumol (**1**), xanthohumol D (**3**) and xanthohumol 4′-*O*-β-D-(4‴-*O*-methyl)-glucopyranoside (**5**), are easily accessible and promising candidates as proapoptotic agents in canine leukemia/lymphoma treatment. Three selected compounds induced apoptotic cell death in three different canine leukemia/lymphoma cell lines, although the mechanism of action of these compounds was somewhat different in each of the cell lines tested. ROS production and changes in mitochondrial potential were not detected in the GL-1 cell line. The downregulation of Bcl-2 was detected especially in cells treated with xanthohumol. Xanthohumol has strong anti-proliferative properties, but it also has disadvantages. It is hardly soluble and shows estrogenic activity. Therefore, new derivatives with better properties are being sought. The most promising derivative is glycosylated compound **5** due to its much better solubility and possibly its bioavailability in the body. The influence of these compounds on healthy canine cells as well as their efficacy in in vivo models can be addressed in future investigations.

## Figures and Tables

**Figure 1 ijms-24-11724-f001:**
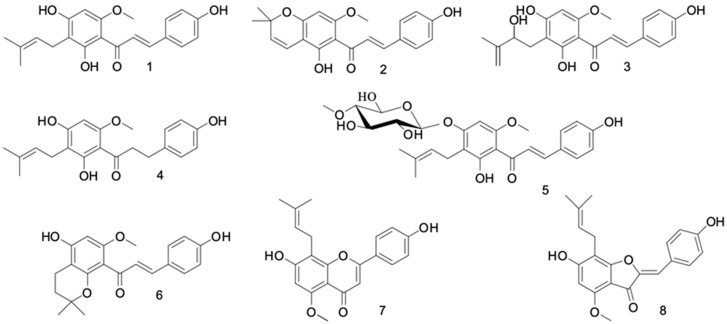
Hops flavonoids (**1**–**4**) and their derivatives obtained by microbial and chemical modifications (**5**–**8**).

**Figure 2 ijms-24-11724-f002:**
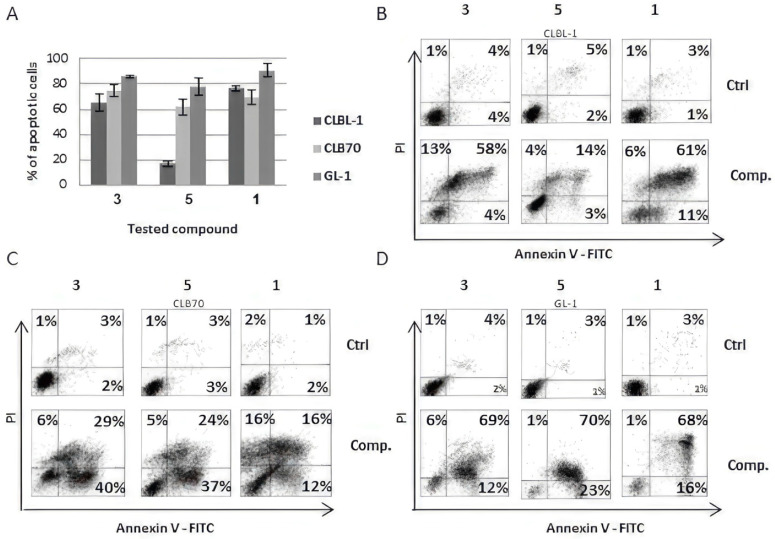
Evaluation of apoptosis induced by xanthohumol and its most active derivatives in CLBL-1, CLB70 and GL-1 (Annexin V-FITC/PI (propidium iodide) staining): (**A**) percentage of apoptotic cells after 48 h exposure to the concentration of 3 µM; ((**B**) CLBL-1 and (**C**) CLB70); and 10 µM ((**D**). GL-1). Different concentrations were used due to the different sensitivities of tested cell lines. Values are expressed as the means ± SD of three independent experiments. **3**—Xanthohumol D, **1**—Xanthohumol, **5**—Xanthohumol 4′-O-β-D-(4‴-O-methyl)-glucopyranoside.

**Figure 3 ijms-24-11724-f003:**
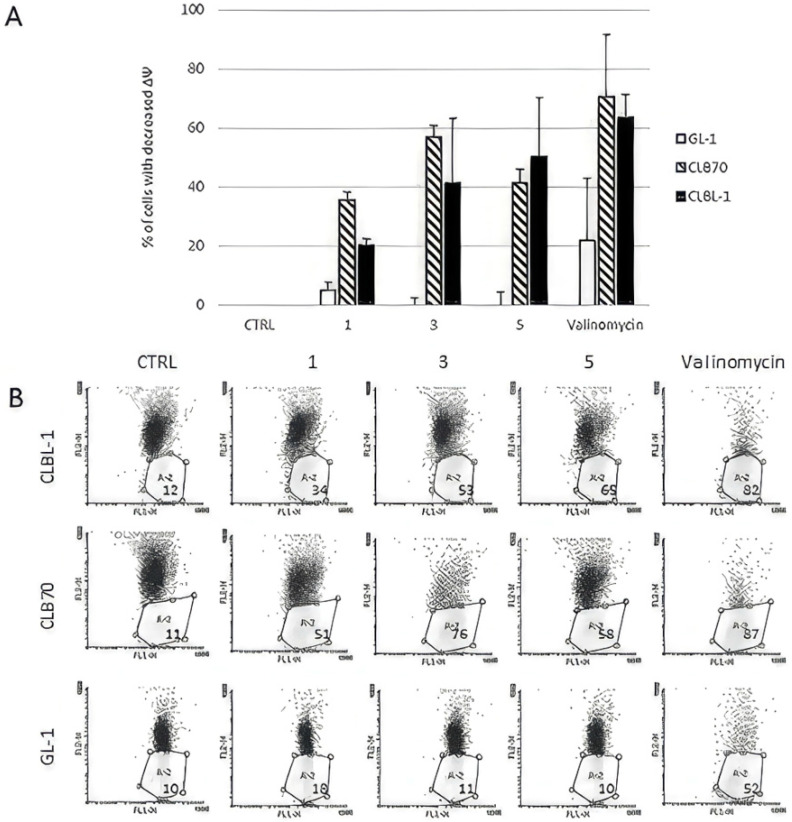
The analysis of mitochondrial potential decrease (ΔΨ) in canine cancer cells. Cells were treated with the investigated flavonoids (concentration 3 μM each) for 24 h, stained with JC-1 dye and the number of cells with decreased ΔΨ was measured: (**A**) the results were normalized to the control values with SD plotted on the graph. **3**—Xanthohumol D, **1**—Xanthohumol, **5**—Xanthohumol 4′-*O*-β-D-(4‴-*O*-methyl)-glucopyranoside; and (**B**) representative dotplots showing the changes in the mitochondrial potential. The R2 field contains the percentage of cells.

**Figure 4 ijms-24-11724-f004:**
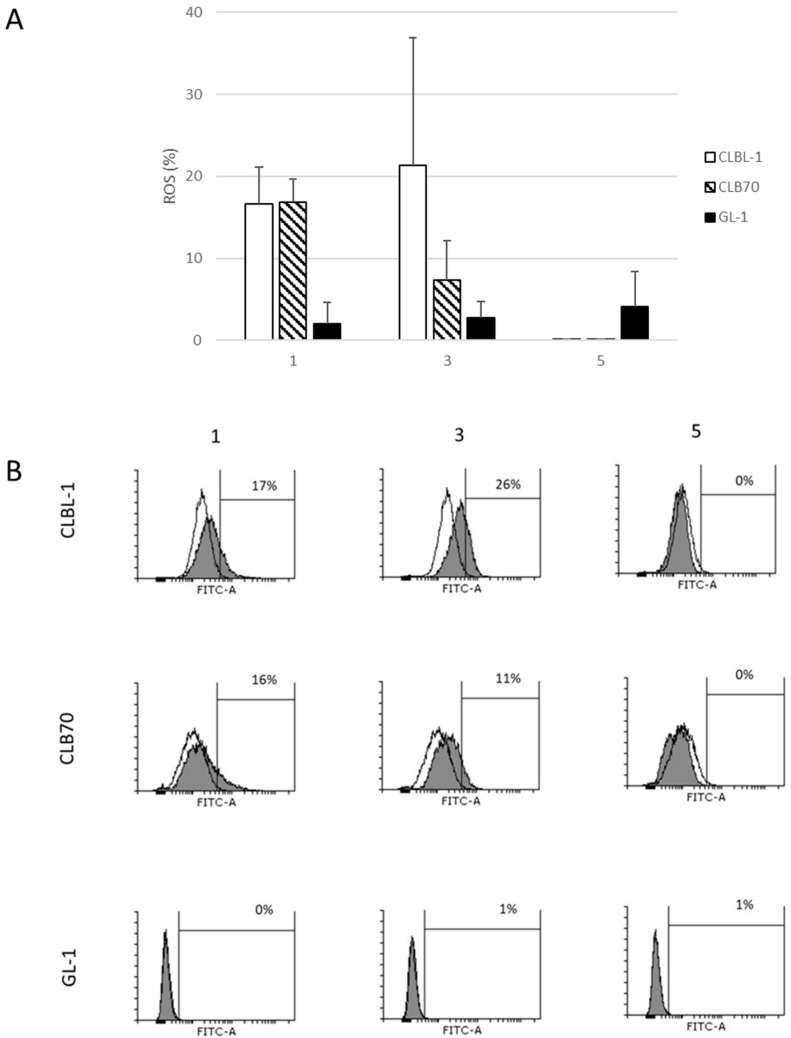
ROS production in canine cancer cells. The cells were incubated with the flavonoid (concentration 10 μM each) for 1 h and with an H2DCFDA reagent for an additional hour and measured using a flow cytometer: (**A**) the fluorescence of reagent was measured in FL-1 (FITC-A) channel. The percentage of ROS production with SD calculated; and (**B**) representative histograms. The graphs show the control cells without the addition of the flavonoid (white field) and the shift generated by the addition of the flavonoid (gray field). **3**—Xanthohumol D, **1**—Xanthohumol, **5**—Xanthohumol 4′-O-β-D-(4‴-O-methyl)-glucopyranoside.

**Figure 5 ijms-24-11724-f005:**
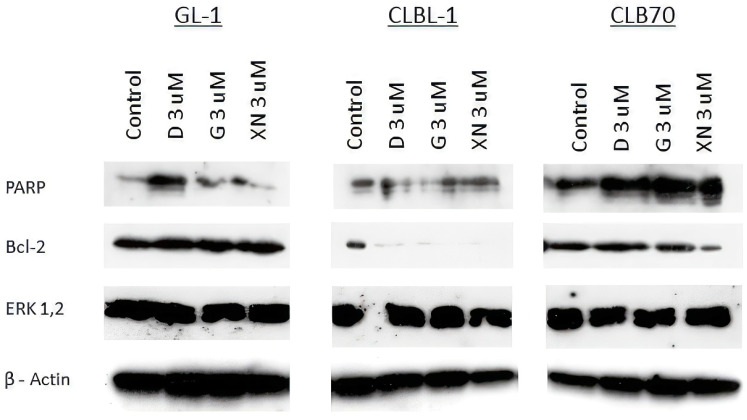
Representative immunoblots of PARP, ERK and Bcl-2 expression in canine cancer cells after 1, 3 and 5 treatments for 48 h. Each compound was used in concentration 3 μM. Actin was used as a loading control. **3**—Xanthohumol D, **1**—Xanthohumol, **5**—Xanthohumol 4′-*O*-β-D-(4‴-*O*-methyl)-glucopyranoside.

**Table 1 ijms-24-11724-t001:** Comparison of concentrations (µM) of xanthohumol and its derivatives that inhibited 50% of the cell viability (IC_50_ values) in CLBL-1, CLB70 and GL-1 cells after 48 h of exposure. Values are expressed as the means ± SD of three independent experiments.

Cell Line	Compound							
	1	2	3	4	5	6	7	8
CLB70	1.35 ± 0.13	6.73 ± 1.36	0.55 ± 0.17	7.42 ± 1.36	1.20 ± 0.34	7.65 ± 2.90	5.88 ± 1.05	8.39 ± 1.44
CLBL-1	1.48 ± 0.26	5.10 ± 1.16	1.08 ± 0.37	5.83 ± 1.77	1.82 ± 0.12	6.91 ± 2.07	3.44 ± 2.40	5.00 ± 1.58
GL-1	6.25 ± 1.15	6.85 ± 1.29	5.27 ± 0.56	13.49 ± 2.46	4.87 ± 1.20	18.70 ± 2.41	9.16 ± 0.21	14.35 ± 3.00

**Table 2 ijms-24-11724-t002:** Retention time (Rt) and wavelength (λ) of the obtained compounds.

Compound	Rt [min]	Detection at λ [nm]
1″,2″-dihydroxanthohumol K (**6**)	1.67	330
xanthohumol 4′-*O*-β-D-(4‴-*O*-methyl)-glucopyranoside (**5**)	2.58	370
2,3-dehydroisoxanthohumol (**7**)	3.00	370
xanthohumol D (**3**)	3.77	370
(*Z*)-6,4′-dihydroxy-4-methoxy-7-prenylaurone (**8**)	3.80	400
α,β-dihydroxanthohumol (**4**)	5.10	280
xanthohumol (**1**)	5.87	370
xanthohumol C (**2**)	6.37	370

## Data Availability

Data are available upon reasonable request from the corresponding author.

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
