# Peer review of "The Effect of Xanthohumol Derivatives on Apoptosis Induction in Canine Lymphoma and Leukemia Cell Lines"

_ijms, 2023, doi:10.3390/ijms241411724_

Round 1
Reviewer 1 Report
Effect of Xanthohumol was evaluated in this manuscript on various hematopoietic cell lines, wherease, several questions should be answered.
1. It seems like Xanthohumol is derived from by-products of beer industry, while its directives were from the process of extraction. If supporting more evidence on why these directives were selected for these in vitro studies. I found several modifications were added to XN but could not tell their general function. This could be important on the hematology field as we dont know much about pharmacology.
2. There are over 40 human B cell lymphoma cell lines, some of which were MCL1 independent that are insensitive to apoptosis. I can not understand why choosing dog lymphoma cell lines. This makes more difficult to conduct in vivo studies in the future.
3. Flow density plots in figure 2 and 3 (R-2) should label percentage of each part. Also, indicate which antibodies/reagents used in FITC channel in figure 4
4. I suggest providing detailed viability data for each compound. It would help explain why you choose 3 uM as working concentration.
5. Figure 5 is over-exposed.
Please reorganize abstract and introduction with focus on relation between Xanthohumol and lymphoma/leukemia.
Author Response
Answers to comments
Reviever 1
- It seems like Xanthohumol is derived from by-products of beer industry, while its directives were from the process of extraction. If supporting more evidence on why these directives were selected for these in vitro studies. I found several modifications were added to XN but could not tell their general function. This could be important on the hematology field as we dont know much about pharmacology.
RESPONSE:
The intention of the authors in selecting the xanthohumol derivatives used in this study was to determine how:
1) the type of flavonoid skeleton (chalcone 1, α,β-dihydrochalcone 4, flavone 7, aurone 8);
2) modification in the prenyl group (cyclization - compounds 2 and 6, hydroxylation - compound 3);
3) the presence of a sugar moiety (compound 5) affect the antiproliferative activities against canine cancer cell lines. To highlight the intention and the goal of the study, the “Discussion” section was supplemented with this information.
- There are over 40 human B cell lymphoma cell lines, some of which were MCL1 independent that are insensitive to apoptosis. I can not understand why choosing dog lymphoma cell lines. This makes more difficult to conduct in vivo studies in the future.
RESPONSE:
Our research groups for several years have been focused on canine cancers, especially on lymphoma and leukemia . B-cell lymphoma is very common in dogs and accounts for about 30% of all cancers. Typical schemes adopted from human medicine like CHOP or COAP are used for their treatment. Introducing new substances to dog therapy is faster and cheaper. There are a number of different options for the treatment of human haematological malignancies.
- Flow density plots in figure 2 and 3 (R-2) should label percentage of each part. Also, indicate which antibodies/reagents used in FITC channel in figure 4
RESPONSE:
We added numeric data to 3 fields, but it was difficult to add to the field with live cells. The percentage content is also placed in the field R2 ( Figure 3). An explanation of the FITC channel is provided in the description of Figure 4.
- I suggest providing detailed viability data for each compound. It would help explain why you choose 3 uM as working concentration.
RESPONSE:
We added a new figure to supplementary file Figure S1 showing the curves for three selected compounds.
The 3 µM concentration was chosen because it reduces cell viability by 80-90% for the two cell lines. The use of higher concentrations may make it impossible to compare the activity of different compounds on different lines. In addition, such concentrations can be achieved in the blood serum after oral administration.
- Figure 5 is over-exposed.
RESPOSE:
New MAPK blot is added. The actin content in the cells is very high and it is difficult to obtain weaker bands.
Comments on the Quality of English Language
Please reorganize abstract and introduction with focus on relation between Xanthohumol and lymphoma/leukemia.
RESPONSE:
Because of word limit in ABSTRACT, we only slightly modified this section. We added a few sentences concerning human and canine lymphoma. Only a few papers describe the effect of xanthohumol on lymphoma/leukemia cells. Some of them is already described in the introduction.
Reviewer 2 Report
In the manuscript “The effect of xanthohumol derivatives on apoptosis induction in canine lymphoma and leukemia cell lines” Dr a GrudzieÅ„ and colleagues investigates predominantly the apoptotic potential in xanthohumol (XH) and their natural and semisynthetic derivatives against various canine leukemia/lymphoma. The study is well written and ambitious in its methodology. The main weakness is the poor support for the statement that the tested compounds would generate low toxicity in normal cells, as this is not shown. This is stated in the end of the conclusion and the authors should be more humble in the general introduction and background concerning this, until better investigated.
Specific:
Introduction
Please add the limitations known for XH – namely low bioavailability and also problems with extraction of the compound from spent hops in the brewery industry.
A figure depicting the entire range of mode of actions for XH and how it may impact cell viability would increase the understanding of choosing the used experiments and rise the interest for the study in general. An example of such a figure could be found in: Jiang C-H et al., Front Pharmacol. 2018; 9: 530 Figure 2.
P 2 second to last section: Please provide references for prevalence of canine lymphoma and the corresponding NHL described to support the statement that canine lymphoma is more common than NHL in man.
P 3 2.1: Please correct ERROR reference not found.
P3 2.2. Please add references to the used cell lines the first time of appearance. They are correctly referenced in the M&M section, but should also be referenced here.
P 4 2.3: Please correct ERROR reference not found.
P4 Figure 2. Please explain abbreviation PI used on Y-axis in B, C and D.
P 5 first row: Please correct ERROR reference not found.
P 5 Figure 3 B: Editorial office needs to decide if figure legends are according to journal instructions. In my view they are too small to ensure visibility. Possibly this figure could be arranged in another way to improve understanding?
P 6 first row: Please correct ERROR reference not found.
P 6 last row: Please correct ERROR reference not found.
P 7 Figure 5. Please expand the discussion on finding in CLBL-1 with BCL-2.
In general acceptable level of English.
Author Response
Reviever 2
Comments and Suggestions for Authors
In the manuscript “The effect of xanthohumol derivatives on apoptosis induction in canine lymphoma and leukemia cell lines” Dr a GrudzieÅ„ and colleagues investigates predominantly the apoptotic potential in xanthohumol (XH) and their natural and semisynthetic derivatives against various canine leukemia/lymphoma. The study is well written and ambitious in its methodology. The main weakness is the poor support for the statement that the tested compounds would generate low toxicity in normal cells, as this is not shown. This is stated in the end of the conclusion and the authors should be more humble in the general introduction and background concerning this, until better investigated.
RESPONSE:
We did not test the effect of XN on normal canine cell lines due to their unavailability. The literature describes the effect of XN and its derivatives on various normal human and mouse cell lines and shows that the inhibitory effect on cancer cell lines is much stronger.
Specific:
Introduction
Please add the limitations known for XH – namely low bioavailability and also problems with extraction of the compound from spent hops in the brewery industry.
RESPONE:
Additional information associated with problems with isolation of the prenylflavonids from spent hops was added in the "Introduction" section.
Data on the pharmacokinetics of xanthohumol have been supplemented in the "Discussion" section, where the bioavailability of xanthohumol in rat studies is given. In addition, the section is expanded by the explanation about the selection of compounds used in the study, including the glycosylated derivative of xanthohumol 5, which, through its better water solubility, could potentially increase its bioavailability.
A figure depicting the entire range of mode of actions for XH and how it may impact cell viability would increase the understanding of choosing the used experiments and rise the interest for the study in general. An example of such a figure could be found in: Jiang C-H et al., Front Pharmacol. 2018; 9: 530 Figure 2.
RESPONSE:
It is difficult to pinpoint the essential mechanism of action because different derivatives show slightly different effects in similar cell lines. Canine cancer research is limited due to the lack of specific antibodies. We have tried to adopt antibodies against various human antigens for Western Blotting but without success.
P 2 second to last section: Please provide references for prevalence of canine lymphoma and the corresponding NHL described to support the statement that canine lymphoma is more common than NHL in man.
RESPONSE:
Literature showing the prevalence of lymphomas in dogs and humans has been added. DLBCL is more common in dogs.
Canine lymphoma: a review.
Zandvliet M.
Vet Q. 2016 Jun;36(2):76-104. doi: 10.1080/01652176.2016.1152633. Epub 2016 Mar 8.
P 3 2.1: Please correct ERROR reference not found.
RESPONSE:
The error has been corrected, the brackets should contain the figure number, not literature references.
P3 2.2. Please add references to the used cell lines the first time of appearance. They are correctly referenced in the M&M section, but should also be referenced here.
RESPONSE:
P 4 2.3: Please correct ERROR reference not found.
RESPONSE:
Error is corrected.
P4 Figure 2. Please explain abbreviation PI used on Y-axis in B, C and D.
RESPONSE:
The explanation of the abbreviation PI is added to the figure description.
P 5 first row: Please correct ERROR reference not found.
Error is corrected.
P 5 Figure 3 B: Editorial office needs to decide if figure legends are according to journal instructions. In my view they are too small to ensure visibility. Possibly this figure could be arranged in another way to improve understanding?
RESPONSE:
We will enlarge the descriptions if necessary.
P 6 first row: Please correct ERROR reference not found.
Error is corrected
P 6 last row: Please correct ERROR reference not found.
Error is corrected
P 7 Figure 5. Please expand the discussion on finding in CLBL-1 with BCL-2.
RESPONSE:
We've added a few sentences to the Discussions.
Round 2
Reviewer 1 Report
I still suggest try on human cell lines in the future as some B cell lymphoma is associated with poor outcome in CHOP.
It has been improved.